# C6 Ceramide Inhibits Canine Mammary Cancer Growth and Metastasis by Targeting EGR3 through JAK1/STAT3 Signaling

**DOI:** 10.3390/ani14030422

**Published:** 2024-01-27

**Authors:** Jiayue Liu, Fangying Zhao, Yan Zhang, Zhaoyan Lin, Ji-Long Chen, Hongxiu Diao

**Affiliations:** 1Joint Laboratory of Animal Pathogen Prevention and Control of Fujian-Nepal, College of Animal Sciences, Fujian Agriculture and Forestry University, Fuzhou 350002, China; shuijing1104744162@163.com (J.L.); mszyzhangyan@foxmail.com (Y.Z.); chenjl@im.ac.cn (J.-L.C.); 2Department of Clinical Veterinary Medicine, College of Veterinary Medicine, China Agricultural University, Beijing 100193, China; zhaofy97@126.com; 3Key Lab for Integrated Chinese Traditional Veterinary Medicine and Animal Healthcare in Fujian Province, College of Animal Sciences, Fujian Agriculture and Forestry University, Fuzhou 350002, China; 000q822065@fafu.edu.cn

**Keywords:** C6 ceramide, anti-tumor activity, EGR3, JAK1/STAT3 signaling, mammary tumor, dog

## Abstract

**Simple Summary:**

Mammary tumors are the most prevalent type of tumor in the canine population. C6 ceramide has been identified as an essential factor in various functions in cancer; however, the role of C6 ceramide in canine mammary cancer remains unknown. Therefore, we investigated the role of C6 ceramide in the progress of canine mammary cancer and explored its potential mechanism. The findings show that C6 ceramide inhibited cell growth, migration, and invasion in vitro. C6 ceramide decreased tumor growth and metastasis in the lungs without side effects in xenograft models. Further exploring found that EGR3 is a target agent of C6 ceramide, promoting canine mammary cancer cell proliferation and migration by activating the pJAK1/pSTAT3 signaling pathway. This study revealed the anti-tumor activity of C6 ceramide in canine mammary cancer both in vivo and in vitro and revealed that EGR3 is a potential biomarker in canine mammary cancer prognosis and treatment.

**Abstract:**

Cancer is the leading cause of death in both humans and companion animals. Canine mammary tumor is an important disease with a high incidence and metastasis rate, and its poor prognosis remains a serious clinical challenge. C6 ceramide is a short-chain sphingolipid metabolite with powerful potential as a tumor suppressor. However, the specific impact of C6 ceramide on canine mammary cancer remains unclear. However, the effects of C6 ceramide in canine mammary cancer are still unclear. Therefore, we investigated the role of C6 ceramide in the progress of canine mammary cancer and explored its potential mechanism. C6 ceramide inhibited cell growth by regulating the cell cycle without involving apoptosis. Additionally, C6 ceramide inhibited the migration and invasion of CHMp cells. In vivo, C6 ceramide decreased tumor growth and metastasis in the lungs without side effects. Further investigation found that the knockdown of EGR3 expression led to a noticeable increase in proliferation and migration by upregulating the expressions of pJAK1 and pSTAT3, thus activating the JAK1/STAT3 signaling pathway. In conclusion, C6 ceramide inhibits canine mammary cancer growth and metastasis by targeting EGR3 through the regulation of the JAK1/STAT3 signaling pathway. This study implicates the mechanisms underlying the anti-tumor activity of C6 ceramide and demonstrates the potential of EGR3 as a novel target for treating canine mammary cancer.

## 1. Introduction

Cancer is one of the prominent causes of death in both humans and companion animals. According to GLOBOCAN, female breast cancer is the most frequently diagnosed cancer globally [1]. Similarly to human breast cancer, canine mammary tumors (CMTs) are the most prevalent type of tumor in the canine population, accounting for approximately 60% of elderly dogs (ranging from 8 to 13 years) [2]. Additionally, about half of these CMTs are malignant [3]. Treatment options for CMTs include surgery, chemotherapy, radiation therapy, and combinations of these treatments, depending on factors such as tumor size, location, and stage of the disease. Surgery is often considered the most effective choice for CMT patients as it involves the removal of affected glands and nearby lymph nodes [4], with exceptions for inflammatory carcinomas and distant metastasis carcinomas. However, it should be noted that 58% of dogs developed a new tumor in the ipsilateral mammary chain [5], highlighting the significant clinical problem that CMTs present. Therefore, to improve CMT management, the main challenge that remains open is finding new therapeutic strategies useful in treating the disease.

Sphingolipids are bioactive molecules in cell membranes that play crucial roles in various signal transduction pathways involving cell death/survival, proliferation, migration, and senescence [6]. As a group of sphingolipid metabolites, short-chain cell-permeable ceramides have been identified as an essential factor in various functions in cancer and exhibit potent anticancer activity either alone or in combination with other known anticancer drugs. C2 ceramide and C6 ceramide have been utilized to mimic an endogenous lipid increase, the apoptosis by NF-κB DNA-binding, caspase-3 activation, poly (ADP-ribose) polymerase degradation, and mitochondrial proapoptotic protein releases have been observed [7,8]. Additionally, several ceramide-based cancer therapeutics are currently being tested in preclinical and clinical (phase I and II) trials for human breast cancer [9]. Naturally occurring canine mammary tumors share similar features with human breast cancer, including live environment, etiology, sub-type classification, signaling pathway activation, and mutational profiles [10,11]. These demonstrate that canines can serve as a comparative and promising model for human mammary cancer research. However, it is still unclear whether C6 ceramide has any impact on the progression of canine mammary tumors.

Early growth response 3 (EGR3) is a member of the C2H2-type zinc-finger transcription factor family. Functionally, EGR3 has been implicated in the suppression of tumor initiation and progression. Recently, dysregulation of EGR3 has been correlated with prognosis and survival. EGR3 was found to be significantly downregulated in tumor tissues compared with non-tumor tissues, such as nasopharyngeal carcinoma [12], prostate cancer [13], hepatocellular carcinoma [14], and epithelial ovarian cancer [15], which results in the poor prognosis and low survival of patients, thus suggesting that EGR3 could potentially serve as a biomarker in the diagnosis and prognosis assessment of cancer. Moreover, EGR3 is identified as a direct target of microRNAs regulating cancer development. An example is seen in the study conducted by Li et al. where miR-483-5p promoted the migration of nasopharyngeal carcinoma cells by targeting EGR3 [16]. More than 90% of cancer-related deaths are caused by metastasis [17]. Several studies demonstrated that EGR3 plays a role in cancer cell migration and invasion, indirectly influencing cancer progression [12,18,19]. However, the specific regulatory effect and molecular aspects of EGR3 in canine mammary cancer remain unknown.

Based on the aforementioned data, we sought to evaluate C6 ceramide as a potential anti-tumor drug in canine mammary tumor cells in vitro and in a xenograft model. Additionally, we sought to underline the role of RGR3 in canine mammary cancer progression.

Finally, we have demonstrated the anti-tumor effects of C6 ceramide in vitro and in vivo on the proliferation and metastasis of canine mammary cancer. Additionally, we have identified the deletion of EGR3 to be a crucial genetic alteration in canine mammary cancer, which regulates the JAK1/STAT3 signaling pathway.

## 2. Materials and Methods

### 2.1. Statement of Ethics

This study received approval for research ethics from the Animal Ethics Committee of China Agricultural University, Beijing, China (AW01303202-2-1), according to the standard ethical guidelines for the Chinese government’s care and use of laboratory animals.

### 2.2. Cell Lines and Cell Culture

The CMT7364 and CHMp-Luciferase-EGFP were kindly provided by the College of Veterinary Medicine at China Agricultural University (Beijing, China). CHMp, CIPp, CIPm, CTBp, and CTBm were kindly provided by the Graduate School of Agricultural and Life Sciences at the University of Tokyo (Tokyo, Japan). The 293T cell line was purchased from ATCC (American Type Culture Collection) (Manassas, VA, USA). All cell lines were cultured in DMEM supplemented with 10% fetal bovine serum (FBS), penicillin (100 U/mL), and streptomycin (100 μg/mL), and they were incubated in a humidified atmosphere with 5% CO_2_ at 37 °C. The investigation of mycoplasma infection was conducted using the MycoBlue Mycoplasma Detector (Vazyme, Nanjing, China) prior to the experiment. Only cells that tested negative were used in this study.

### 2.3. Cell Viability Assay

The proliferation was evaluated using a Cell Counting Kit-8 (CCK-8) (Beyotime Institute of Biotechnology, Shanghai, China). All canine mammary tumor cells were plated in 96-well plates at 1 × 10^4^ cells/well and incubated overnight to allow for attachment. Cells were treated with different concentrations of C6 ceramide (d18:1/6:0) (Avanti Polar Lipids, Birmingham, AL, USA) for 72 h. CCK-8 (10 μL) was added to each well and, following 1 h of incubation, the optical density was read at 450 nm with a microplate reader (Tecan Austria GmbH Untersberg, Grödig, Austria). Each group had five biological replicates.

For the colony formation assay, 200 cells/well were plated in 6-well plates and incubated overnight to allow for attachment. A total of 8 μM of C6 ceramide was incubated for 24 and 48 h. After treatment, the plates were washed and cultured with 10% FBS DMEM for 8–10 days. The attached cells were then stained with 0.1% (*w*/*v*) crystal violet, and the wells were photographed. Each group had three biological replicates.

### 2.4. Observations of Cell Morphology and Migration

Cells in single-cell suspension were plated in 6-well plates at 2 × 10^5^ cells/well and incubated overnight to allow for attachment. A wound was created with a 200 μL pipette tip. After washing two times with PBS to remove cell fragments, cells with different concentrations of C6 ceramide were cultured with 2% FBS DMEM. Then, images of the wounds and cells were photographed using a bright-field microscope (CKX41; Olympus, Tokyo, Japan) to evaluate cell migration and observe cell morphology. Image J 1.53c (National Institutes of Health, Bethesda, MA, USA) was used to analyze the migration rate. Each group had three biological replicates.

### 2.5. Cell Migration Assay

Cell migration was determined using the wound healing assay. Briefly, 2 × 10^5^ cells/well were plated in 6-well plates and incubated overnight to allow for attachment. A wound was created with a 200 μL pipette tip. After washing two times with PBS to remove cell fragments, cells with different groups were cultured with 2% FBS DMEM. Then, images of the wounds were photographed to evaluate cell migration. Image J 1.53c (National Institutes of Health, Bethesda, MA, USA) was used to analyze the migration rate. Each group had three biological replicates.

### 2.6. Invasion Assay

Matrigel^TM^ (BD Biosciences, San Jose, CA, USA) was mixed with DMEM at a ratio of 1:30. Then, Matrigel (100 μL) was added to the upper chamber of the 24-well transwell filters for 1 h at 37 °C for the coated membrane. A total of 2000 cells/well were seeded in the upper chamber with 100 μL of FBS-free DMEM in different groups, and the lower chambers were filled with 500 μL of complete DMEM with 10% FBS. Invading cells were fixed with 4% paraformaldehyde, stained with 0.1% (*w*/*v*) crystal violet, photographed using a microscope (CKX41; Olympus), and then counted using. Image J 1.53c (National Institutes of Health, Bethesda, MA, USA). Each group had three biological replicates.

### 2.7. Flow Cytometry

A total of 2 × 10^5^ cells/well were plated in 6-well plates and incubated overnight to allow for attachment. After 8 μM of C6 ceramide was incubated for 48 h, cell apoptosis was detected through an annexin V-FITC/propidium iodide (PI) detection kit (BD, USA) (BD Biosciences, San Jose, CA, USA) following the manufacturer’s instructions. Cell cycle fractions were detected through the PI/RNase staining buffer (BD Biosciences, San Jose, CA, USA). All samples were analyzed on a MoFlo XDP flow cytometer (Beckman Coulter, Brea, CA, USA), and data were analyzed through Summit software (Version 5.5, North Little Rock, AR, USA).

### 2.8. Mouse Xenografts

Tumor xenografts were established in 5-week-old female BALB/c nude mice (Vital River, Beijing, China). For primary tumor models, 1 × 10^7^ CHMp-Luciferase-EGFP cells were resuspended in PBS (100 μL) and injected subcutaneously into the left mammary fat pad. Once the tumor volume reached 50 mm^3^, treatment with C6 ceramide would begin and last two weeks. Briefly, 30 mg/kg or 60 mg/kg of C6 ceramide was injected intraperitoneally daily for 15 days, and 2 mg/kg of cisplatin (CDDP) was used as a positive control. For tumor metastasis models, 1 × 10^5^ CHMp-Luciferase-EGFP cells were resuspended in PBS (100 μL) and injected intravenously into the tail vein. On the second day of injection, treatment with C6 ceramide was the same as in primary models. The weight and volume of the tumors were monitored every day. Tumor volume was calculated using the formula as follows: length × width^2^/2. At 1, 5, 10, and 15 days after implantation, mice were injected with D-luciferin potassium salt (150 mg/kg, intraperitoneal). After 10 min, luminescence and EGFP in mice were visualized using the IVIS imaging system (PerkinElmer Inc., Waltham, MA, USA) through anesthesia with isoflurane (HFQ Biotechnology, Nantong Jiangsu, China) as performed in the previous study [20]. At the end of the experiment, mice were euthanized through isoflurane and CO_2_. Tumor tissues, lungs, and spleens were collected to measure weights or volumes.

### 2.9. RNA Interference and Transfection

Short hairpin RNA (shRNA) was designed using BLOCK-iT™ RNAi Designer for knockdown of EGR3. The sh-RNA sequences were as follows: 5′-GCAACAAGACCGTGACCTACT-3′ and 3′-AGTAGGTCACGGTCTTGTTGC-5′. DNA transfection was performed using Lipo8000TM (Beyotime, Shanghai, China). Lentiviruses expressing shRNAs were packaged in 293T cells. The shRNA constructs and lentiviral packaging plasmids (PLP1, PLP2, and PLP-VSVG) were co-transfected into these cells for 48 h. Then, the virus-containing supernatant was passed through a 0.22 μm filter to eliminate cells. A total of 2 × 10^5^ cells/well in 6-well plates were infected with the virus in a medium containing 5 μg/mL of polybrene (Beyotime, Shanghai, China) using a spin infection technique through centrifugation at 2100 rpm for 2 h. Transduction efficiency was determined through fluorescence microscopy.

### 2.10. Reverse Transcription PCR (RT-PCR) and Quantitative Real-Time PCR (qPCR)

According to the manufacturer’s instructions, RNAs were isolated using the Cell Total RNA Isolation Kit (Foregene, Chengdu, China). Reverse transcription was using TransScript^®^ One-Step RT-PCR SuperMix (TransGen Biotech, Beijing, China). The quantitative real-time PCR used was TransStart^®^ Green qPCR SuperMix (TransGen Biotech, Beijing, China). Results were expressed using the 2^−∆∆Ct^ comparative method. GAPDH was used as the reference gene. The primers were designed through Primer 5.0 software. The sense and antisense sequences of EGR3 were 5′-ACGGTAGAGGCTTCGGTTC-3′ and 3′-TAAGAGAGTTCCGGGTTGGG-5′; the sense and antisense sequences of GAPDH were 5’-TACCACCATGTACCCTGGCA-3′ and 3′-CTTCTGGGTGGCAGTGAT-5′.

### 2.11. Western Blotting

Cells in single-cell suspension were plated in 6-well plates at 2 × 10^5^ cells/well and incubated overnight to allow for attachment. After 8 μM of C6 ceramide treatment for 48 h, cells were lysed with RIPA buffer (Beyotime, Shanghai, China) and supplemented with PMSF (Beyotime, Shanghai, China) on ice for 20 min. Then, proteins were quantified using the BCA protein assay kit (Beyotime, Shanghai, China). SDS-PAGE separated the total protein (20 μg) on a 10% gel, which was then transferred onto a nitrocellulose membrane (Millipore, St. Louis, MO, USA) and incubated with primary antibodies EGR3 (bs-6448R, Bioss, Beijing, China, 1:1000) and β-actin (8115-1-RR, Proteintech, Wuhan, China, 1:2000) overnight at 4 °C. HRP-conjugated anti-rabbit (SA00001-2, Proteintech, China, 1:2000) was incubated for 1 h at room temperature. Finally, the membranes were visualized using a chemiluminescence imaging analysis system (Tanon 5200, Shanghai, China). Immunoblotting signals were quantified through densitometry using. Image J 1.53c (National Institutes of Health, Bethesda, MA, USA)

### 2.12. Statistical Analysis

All data are presented as mean ± standard deviation (SD). Statistical significance was determined through the unpaired Student’s *t*-test or two-way ANOVA using GraphPad Prism software (Version 8.0, San Diego, CA, USA) or SPSS18.0 (Version 20, Chicago, IL, USA). * *p* < 0.05, ** *p* < 0.01, and *** *p* < 0.001 were considered significantly different.

## 3. Results

### 3.1. C6 Ceramide Inhibited Canine Cancer Cell Growth by Arresting the Cell Cycle at the S Phase

To assess cell viability and the cytotoxic effects of C6 ceramide, canine mammary cancer cell lines were cultured with various concentrations of C6 ceramide for 72 h, and cell viability was measured using the CCK-8 assay. Treatment with C6 ceramide decreased the cell viability in all cell lines in a dose-dependent manner (Figure 1A). The IC50 values of CIPm, CIPp, CHMp, CMT7364, CTBp, and CTBm were found to be 20.81 ± 5.93 μM, 12.58 ± 3.99 μM, 5.97 ± 1.68 μM, 7.16 ± 2.50 μM, 1.71 ± 0.51 μM, and 1.76 ± 0.63 μM, respectively (Figure 1B). Based on their tumorigenicity, CHMp and CMT7364 were selected for further experiments. The morphological features of the cell lines changed after incubation with C6 ceramide. Cells became round, forming clusters and losing their reticular distribution with a decrease in cell adhesion (Figure 1C). As for the colony formation assay, C6 ceramide reduced the formation of colonies in these two mammary cancer cell lines at both 24 and 48 h (Figure 1D).

To further explore the mechanisms involved in the anti-proliferative effect of C6 ceramide, apoptosis and cell cycle distribution were measured using flow cytometry. Interestingly, no effect on apoptosis was observed in either CMT7364 or CHMp after treatment with 8 μM of C6 ceramide for 48 h. However, C6 ceramide treatment arrested both canine cancer cell lines in the S and G2/M phases of the cell cycle (Figure 1E,F). These findings suggest that C6 ceramide exerts an anti-proliferative effect on canine mammary cancer cells by arresting the cell cycle at the S phase without involving apoptosis.

### 3.2. C6 Ceramide Inhibited Cell Migration and Invasion

The wound assay demonstrated a significantly higher inhibitory effect in both cells that were treated with 8 μM of C6 ceramide for 48 h compared and those with non-treated cells (Figure 2A–D). In the control group, the wounds in CMT7364 and CHMp cells were almost completely healed, with healing percentages of 97.80 ± 0.04% and 84.79 ± 6.65%, respectively. However, the C6 ceramide treatment resulted in a decrease in healing, with percentages of 67.82 ± 15.08% in CMT7364 cells and 40.41 ± 11.21% in CHMp cells after 48 h (Figure 2B,D). In the transwell assay, the number of invading cells was significantly reduced in the C6 ceramide treatment group (Figure 2E,F). These results demonstrate that C6 ceramide can effectively inhibit metastasis in canine mammary cancer cells.

### 3.3. C6 Ceramide Suppresses CHMp Xenograft Tumor Growth In Vivo

For the primary tumor model, the luminescent intensity was significantly stronger in the control group than that in C6 ceramide or CDDP groups at the end of treatment (Figure 3A,B), which was consistent with tumor volume (Figure 3C). The anti-tumor effects of 60 mg/kg of C6 ceramide and 2 mg/kg of CDDP in vivo were similar. However, severe side effects of CDDP were observed during the treatment, like that of losing weight (Figure 3D), and this is why the mice were euthanized at 15 days. Tumors were collected at the end of treatment (Figure 2E,F), the tumor weights decreased in the C6 ceramide treatment, and a significant difference was observed between the 60 mg/kg of C6 ceramide treatment and the control group (Figure 2F). Spleen weights significantly decreased only in the CDDP group; no difference was observed among the control and C6 ceramide treatment groups (Figure 3G,H).

Similar results were found in metastasis models. The luminescent intensity was significantly stronger in the control group than in other groups at the end of treatment, and the inhibitory growth of C6 ceramide and CDDP were equivalent (Figure 4A,B). The weight of the mice continuously decreased in the CDDP treatment, which was not found in the C6 ceramide treatment (Figure 4C). The tumor metastasis of the lung led to a weight increase in the lung, and the treatment with C6 ceramide reversed this phenomenon (Figure 4D,E). As for spleen weight, which had the same trend as that in the primary models, there was a significant decrease in the CDDP group (Figure 4F,G). These results demonstrate that C6 ceramide effectively suppresses the growth of CHMp xenograft tumors in vivo whether in a primary or metastasis model.

### 3.4. Identification of EGR3 as a Target Gene for C6 Ceramide and EGR3 Promoted the Cell’s Abilities of Proliferation and Migration by Regulating the JAK1/STAT3 Pathway

EGR3 plays a crucial role in cancer growth and metastasis. To determine whether the anti-tumor effects of C6 ceramide are correlated with EGR3, we conducted qPCR and Western blotting analyses to confirm this hypothesis. EGR3 expression was significantly induced after treatment with C6 ceramide both at the mRNA and protein levels (Figure 5A–C). To further investigate the functional significance of EGR3 in canine cancer, CHMp cells were generated using shRNAs designed to specifically target the EGR3 gene (sh-EGR3) to reduce the expression of EGR3. Luciferase was used as a control (sh-luc). Substantial decreases in EGR3 expression in CHMp cells with EGR3 knockdown were observed (Figure 5D–F). 

To examine the role of EGR3 in canine cancer, we assessed the impact of the altered EGR3 expression on CHMp cell survival. The knockdown of EGR3 expression in CHMp cells led to a noticeable increase in the proliferation curve starting from the third day (Figure 5G). Likewise, we observed a higher colony formation in CHMp cells expressing sh-EGR3 compared with those expressing sh-luc (Figure 5H). In the wound healing assay, we also observed a significant increase in migration between control and EGR3 knockdown cells at 24 h and 48 h (Figure 5I,J). These findings indicate that silencing EGR3 enhances CHMp cell survival and migration. Next, we investigated whether EGR3 promotes canine tumorigenesis by regulating the JAK1/STAT3 pathway. We examined the expression of JAK1, pJAK1, STAT3, and pSTAT3 in CHMp cells expressing sh-luc and sh-EGR3. As depicted in Figure 5K,L, we detected significantly higher levels of pJAK1 and pSTAT3 in CHMp cells expressing sh-EGR3 compared with the control. At the same time, no significant difference in the expressions of JAK1 and STAT3 was observed between control and EGR3 knockdown cells. These results demonstrate that silencing EGR3 promotes canine cancer cell survival and migration through the involvement of the JAK1/STAT3 pathway.

## 4. Discussion

Canine mammary cancer has always remained a big challenge for veterinary clinicians due to its high incidence and metastasis rate. Our primary aim is to investigate the potent anticancer activity of C6 ceramide in canine mammary cancer and explore its mechanism. C6 ceramide effectively inhibits the proliferation of canine mammary cancer cells by regulating the cell cycle and not inducing apoptosis (Figure 1). There is a subject of controversy regarding the growth inhibition of C6 ceramide in cancer cells through the apoptotic pathway. Several studies have discovered C6 ceramide-induced apoptosis by evaluating cleaved caspase 3 and/or PARP, such as those on hepatocellular cancer [21], cutaneous T cell lymphoma [22], and pancreatic cancer [23]. On the contrary, studies on head and neck squamous cell carcinoma, melanoma, and breast cancer have revealed that the proliferative inhibitory effect of C6 ceramide is not associated with apoptosis [24,25]. This may be related to the diverse metabolic pathways of C6 ceramide in various cancers considering that each type of cancer possesses unique biological characteristics and proliferation mechanisms. To determine the anticancer effects of C6 ceremide on tumor growth in vivo, tumor inhibition assays were first performed as primary xenograft models. As shown in Figure 3, C6 ceramide-treated xenografts showed significant growth suppression, which was consistent with previous studies [26,27,28].

Metastasis to distant organs is the leading cause of over 90% of cancer-related deaths [17]. Unlike primary tumors, metastasis is a systemic disease that cannot be treated with surgical resection or radiation therapy. In the current study, we chose a canine inflammatory mammary carcinoma cell line with high metastasis potential to determine the effects of C6 ceramide on cancer metastasis. A significant difference in the migration and invasion ability of CHMp cells was observed after treatment with C6 ceramide (Figure 2). Consistent with the in vitro data, C6 ceramide suppresses the metastasis rate in nude mice without any difference compared with the CDDP group but no significant weight loss (Figure 4). A study conducted by Liu et al. confirmed that C6 ceramide in multiple myeloma exosomes mediates miR-29b expression, which in turn participates in migration and angiogenesis [29]. Similarly, C6 ceramide treatment resulted in a significant reduction in melanoma and breast cancer migration and metastasis depending on PI3K and PKCζ activation [25]. Combining the data in vivo, it is worth noting that C6 ceramide does not exhibit visible toxic side effects in comparison with the commonly used chemotherapy drug CDDP, and the mice remain in a good mental state and exhibit regular body weight growth. As is known, chemotherapy is one of the main treatments of cancer with significant side effects, such as immune suppression and vomiting, which limit its effectiveness and usage in cancer treatment. However, C6 ceramide has been shown to effectively suppress cancer without side effects in this study, making it potentially advantageous as an anti-cancer agent. 

EGR3 is a zinc-finger transcription factor that has previously been implicated in T cell immunity [30], inflammation [31], and angiogenesis [32], as supported by the literature. Immune surveillance mediated by T cells is crucial in regulating tumor growth [33], and EGR3 plays a dynamic role in regulating the proliferation and survival of effector T cells while controlling excessive inflammation [34,35]. This indicates that EGR3 participates in tumor initiation and progression through its involvement in inflammatory effector pathways. Kwon et al. have demonstrated that EGR3 can bind to the promoter sequences of HDAC6, resulting in the altered expression of interleukin-27 (IL-27), and the EGR3-HDAC6-IL-27 axis is necessary for enhancing the tumorigenic potential of cancer cells in the context of allergic inflammation by mediating cellular interactions [31]. CHMp is a canine inflammatory mammary carcinoma cell line established from a 12-year-old mixed female dog with T4N1 at a clinical stage [36]. Our data revealed the upregulation of EGR3 after C6 ceramide treatment (Figure 5A–C). The lentivirus-mediated EGR3 knockdown in CHMp cells showed higher tumorigenicity, as evidenced by significant increases in cancer cell growth and migration (Figure 5G–J). Inhibition of CHMp cell proliferation and migration by targeting EGR3 with C6 ceramide may also involve inflammation or immunity. Further studies are needed to address this speculation. Additionally, the alteration in the expression of EGR3 has been associated with survival rate and time to first treatment (TTFT) [13,18,37], thus suggesting that EGR3 can serve as a prognostic biomarker in various types of cancer.

The JAK/STAT signaling pathway plays a key role in cancer development, immunity, tumorigenesis, chemotherapy resistance, and metastatic capacity [38,39], and it is associated with poor clinical outcomes when activated [40]. However, the function of EGR3 in modulating the activity of the JAK/STAT3 pathway is not well understood. Our results revealed that the knockdown of the expression of EGR3 promoted the expressions of pJAK1 and pSTAT3, while no differences were observed in the expressions of JAK1 and STAT3 (Figure 5K,L). The activation of JAK1 leads to the phosphorylation of STAT3, resulting in the nuclear translocation of pSTAT3, which in turn activates the expression of downstream genes that regulate cellular progression [38]. Accumulated evidence has clearly shown that activation of STAT3 facilitates cancer invasion and metastases and induces cancer-promoting inflammation, thus inhibiting immune response [41,42]. Moreover, STAT3 is often associated with a worse prognosis and accelerated disease progression [43]. Therefore, we initially discovered that EGR3 can regulate the activation of the JAK1/STAT3 pathway to promote the progression of canine mammary cancer.

As a whole, our findings provide strong evidence that C6 ceramide inhibits the growth and metastasis of canine mammary cancer both in vitro and in vivo. Meanwhile, our research has unveiled the targeting agent EGR3 of C6 ceramide, which functions as a tumor suppressor gene by regulating the JAK1/STAT3 signaling pathway. These findings highlight the potential of using C6 ceramide and targeting EGR3 as a means of preventing or treating canine mammary cancer. Of course, this study has certain limitations. First, how the EGR3 regulates the JAK1/STAT3 pathway is still unclear, and exploring its interacting proteins may help elucidate its functional mechanism. Second, the expression of EGR3 in canine mammary cancer patients is unknown. Further study is required to investigate the expression of EGR3 in clinical patients and analyze its correlation with tumor histological grading and prognosis. This will assist in determining its potential as a biomarker.

## 5. Conclusions

This study demonstrated that C6 ceramide can inhibit the growth and metastasis of canine mammary cancer both in vivo and in vitro. Additionally, it reveals that the target gene of C6 ceramide, EGR3, can effectively exert anti-tumor effects by regulating the JAK1/STAT3 signaling pathway. EGR3 may prove to be a valuable target for diagnosing and treating canine mammary cancer. Future research should focus on investigating the biological implications and prognostic value of EGR3 in canine mammary cancer patients.

## Figures and Tables

**Figure 1 animals-14-00422-f001:**
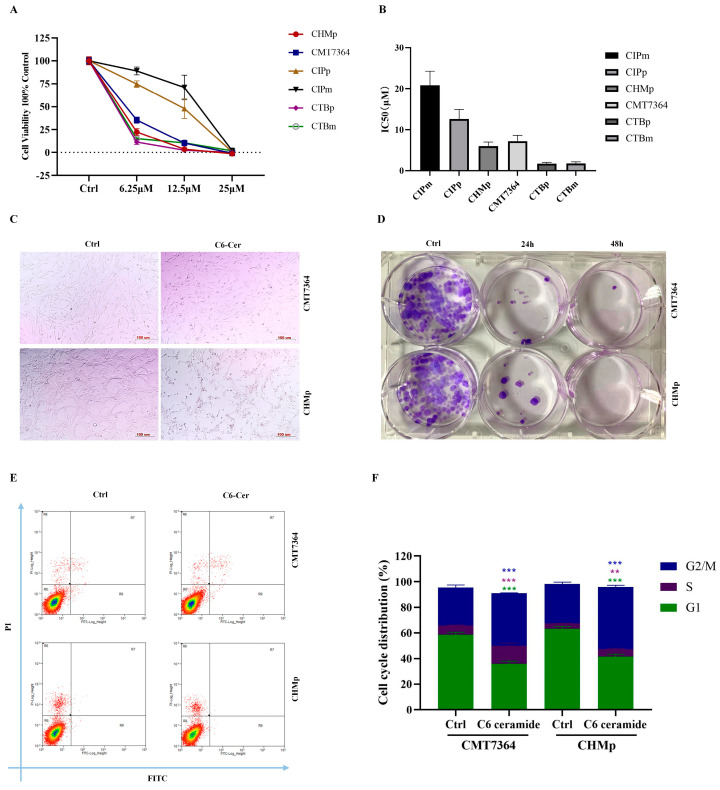
C6 ceramide inhibited the growth of canine mammary cancer cells. (**A**) Cell viability was analyzed using CCK-8 after exposure to C6 ceramide. (**B**) IC_50_ of C6-ceramide on canine mammary cancer cells. (**C**) The morphological features of CMT7364 and CHMp treated with C6 ceramide. (**D**) Colony formation of CMT7364 and CHMp cells treated with C6 ceramide. (**E**) Flow cytometric analysis of annexin V/PI staining in CMT7364 and CHMp cells treated with C6 ceramide. (**F**) Cell cycle distribution was analyzed in CMT7364 and CHMp cells treated with C6 ceramide through flow cytometry. The data are presented as the mean ± SD. ** *p* < 0.01, *** *p* < 0.001.

**Figure 2 animals-14-00422-f002:**
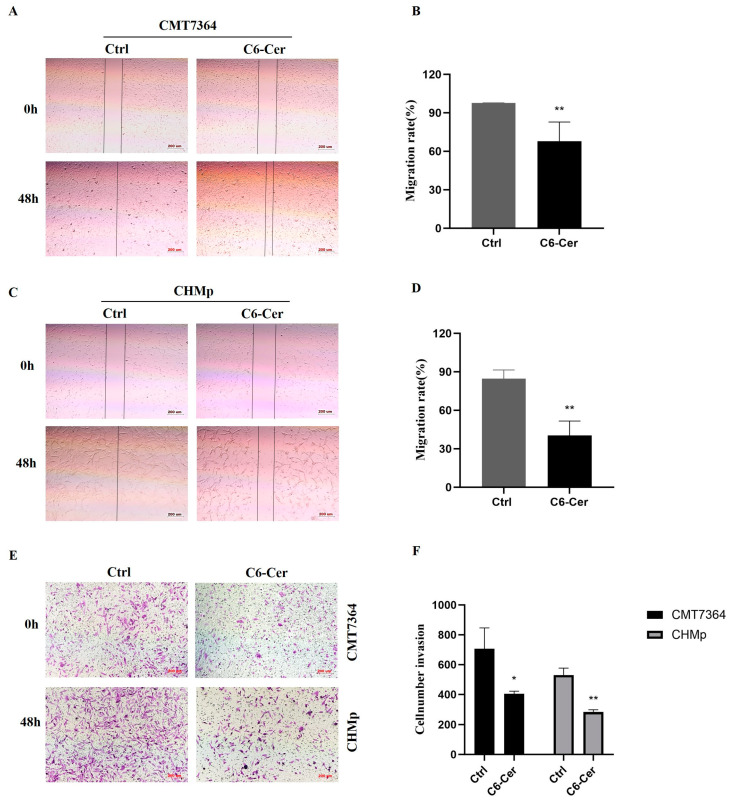
C6 ceramide inhibited migration and invasion of canine mammary cancer cells. (**A**) The migratory ability of CMT7364 was tested at 48 h post-treatment. Scale bar, 200 μm. (**B**) The migratory ability of CMT7364 cells was calculated through Image J. (**C**) The migratory ability of CHMp was tested at 48 h post-treatment. Scale bar, 200 μm. (**D**) The calculation of the migration rate in (**C**) through Image J. (**E**) Invading cells were stained with 0.1% (*w*/*v*) crystal violet. Scale bar, 200 μm. (**F**) The number of invading cells was calculated through Image J. The data were analyzed from three independent experiments. Significance was determined using an unpaired Student’s *t*-test, and the data are presented as the mean ± SD. * *p* < 0.05, ** *p* < 0.01.

**Figure 3 animals-14-00422-f003:**
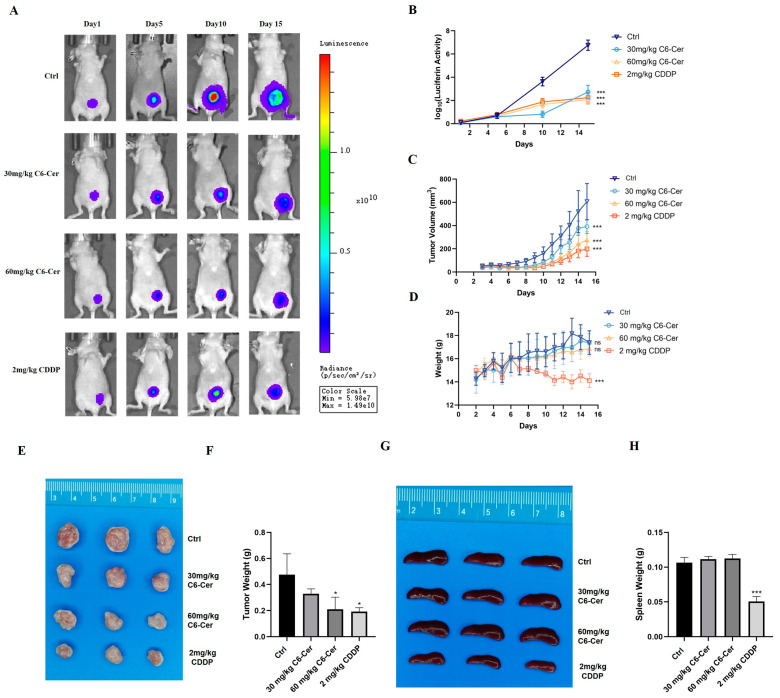
C6 ceramide inhibited the CHMp primary tumor’s model growth. (**A**) Bioluminescence imaging of primary tumor-bearing mice. (**B**) The changes in luminescent intensity in primary tumor-bearing mice. (**C**) Tumor volume. (**D**) Mice body weights. (**E**) Images of tumors at the end of the experiment. (**F**) Tumor weights at the end of the experiment. (**G**) Images of the spleen at the end of the experiment. (**H**) Spleen weights at the end of the experiment. Significance was determined using two-way ANOVA. The data are presented as the mean ± SD. ns, non-significant, * *p* < 0.05, *** *p* < 0.001.

**Figure 4 animals-14-00422-f004:**
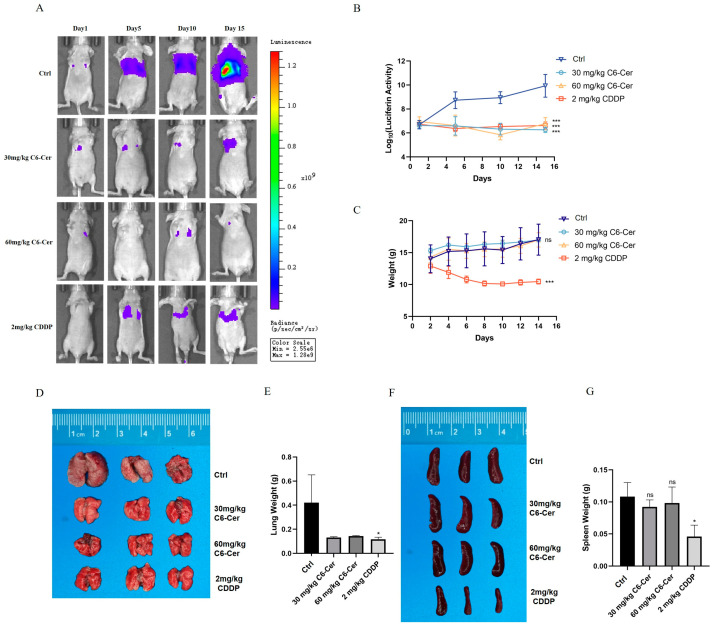
C6 ceramide inhibited the CHMp metastasis tumor’s model growth. (**A**) Bioluminescence imaging of metastasis tumor-bearing mice. (**B**) The changes in luminescent intensity in metastasis tumor-bearing mice. (**C**) Mice body weights. (**D**) Images of the lung at the end of the experiment. (**E**) Lung weights at the end of the experiment. (**F**) Images of the spleen at the end of the experiment. (**G**) Spleen weights at the end of the experiment. Significance was determined using two-way ANOVA. The data are presented as the mean ± SD. ns, non-significant, * *p* < 0.05, *** *p* < 0.001.

**Figure 5 animals-14-00422-f005:**
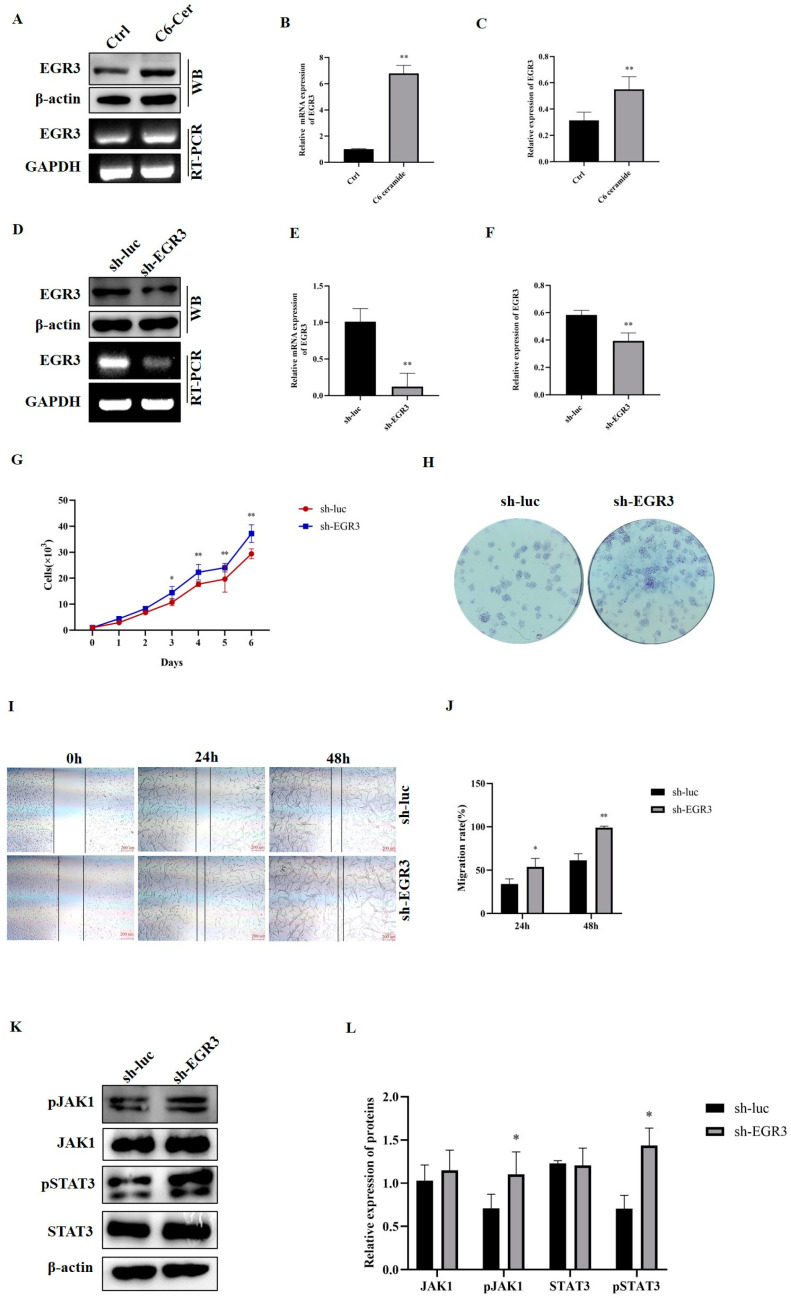
The effects of EGR3 on cell proliferation, migration, and the JAK1/STAT3 pathway. (**A**) RT-PCR and Western blotting were performed to examine the expression of EGR3 in CHMp cells after exposure to C6 ceramide. (**B**) qPCR was performed to examine the EGR3 mRNA level after exposure to C6 ceramide. (**C**) Quantitative analysis of the relative gray of EGR3 in the protein level. (**D**) RT-PCR and Western blotting were performed to examine the expression of EGR3 in CHMp cells expressing control (sh-luc) or EGR3 shRNA (sh-EGR3). (**E**) qPCR was performed to examine the EGR3 mRNA level in CHMp cells expressing sh-luc or sh-EGR3. (**F**) Quantitative analysis of the relative gray of EGR3 in the protein level in CHMp cells expressing sh-luc or sh-EGR3. (**G**) Growth curves of CHMp cells expressing sh-luc or sh-EGR3. (**H**) Colony formation of CHMp cells expressing sh-luc or sh-EGR3. (**I**) The migratory ability of CHMp cells expressing sh-luc or sh-EGR3. Scale bar, 200 μm. (**J**) The calculation of the migration rate in (**C**) throguh Image J. (**K**) The expressions of pJAK1, JAK1, pSTAT3, and STAT3 in CHMp cells expressing sh-luc or sh-EGR3. (**L**) Quantitative analysis of the relative expression of (**E**). Significance was determined using an unpaired Student’s *t*-test, and the data are presented as the mean ± SD. * *p* < 0.05, ** *p* < 0.01. The original images of Western blot are included in the Appendix A.

## Data Availability

The data presented in this study are available on request from the corresponding author. The data are not publicly available due to the data are part of an ongoing study.

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
