# Peer review of "C6 Ceramide Inhibits Canine Mammary Cancer Growth and Metastasis by Targeting EGR3 through JAK1/STAT3 Signaling"

_animals, 2024, doi:10.3390/ani14030422_

Round 1
Reviewer 1 Report
Comments and Suggestions for Authors
The authors examined the in vitro and in vivo effect of C6 ceramide and EGR3 downregulation on canine mammary cancer. C6 ceramide reduced proliferation, colony formation, migration and invasion, without inducing apoptosis in two canine mammary cancer cell lines. In vivo, it reduced tumor growth comparably with cisplatin by causing less side effects. C6 ceramide led to EGR3 upregulation.
EGR knockdown in one of the cell lines increased proliferation, migration and colony formation and activated the JAK/STAT3 pathway.
The authors applied various methods to strengthen the effect of C6 ceramide. However the assumption in the abstract and in the discussion, that EGR3 might be a direct target of C6 ceramide is in my opinion not proven. The authors mention previous RNA-Seq results, but why are they not included in the present study? If they are previously published, they should be cited correctly. Another method to underline this assumption would be to treat the knockdown cells with C6 ceramide. EGR3 upregulation is discussed and proven to lead to higher proliferation. C6 ceramide also upregulates EGR3. Therefore, the knockdown cells should be even more susceptive. Furthermore, in the introduction, EGR3 was described to be downregulated in many tumors. Do the authors assume, that C6 ceramide would not be effective in these kinds of tumors?
The methods are not adequately described in many aspects. For example, the cells were seeded in presence of C6 ceramide, instead of granting the cells a time to attach and equilibrate. How do the authors make sure C6 ceramide does not affect attachment? Was the C6 ceramide dissolved in culture medium? Was it initially dissolved in DMSO or other solvent and is there a solvent control? For many methods, treatment time was described in the results instead of in the methods part. Were the non-invasive cells in the Matrigel Assay removed by cotton swap? How exactly were the mice treated, dosage, solvent, application, treatment interval etc?
In the results, there is a leftover passage from the animals template at the beginning.
Line 217: The effect on cell viability is described as time-dependent. Did the authors also perform CCK-8 assays after 24 and 48 hours?
Line 226: The colony formation test in the text was described as performed after 48 hours, which itself belongs into the methods section, but Fig. 1B shows effects after 24 and 48 hours.
Figure 1: There are no significances in any of the figures, although significances are indicated in the figure legend. In all microscopy images, scale bars are missing and the quality is not appropriate. This also applies to Figure 2A, C and E. Figure 1E: The axis titles are too small, at least apoptotic or dead cells/ FITC and PI channel should be readable. Figure 3B: The axis is semi logarithmic, I would adjust it to log-scale as done in Figure 4B.
Section 3.3: Information about the treatment of mice belongs into the methods section.
Section 3.4: The RNA sequencing results should be included into this manuscript, or at least mentioned in the introduction and methods part as preliminary results or submitted for publication. If the results are not included in this manuscript, RNA-Seq should be removed from the abstract.
A very important result, which is discussed later, is the toxicology in the mouse model. Information about tolerance, observed or not observed side effects etc. should be included in the results section. Furthermore, is there any data about toxicology in dogs or humans?
Comments on the Quality of English LanguageThe overall English is fine, but there are some minor language issues. The article would benefit from a proofread of a native speaker.
Reviewer 2 Report
Comments and Suggestions for Authors
The authors present a study on the effects of C6 ceramide on canine mammary cancer, in a cell line, and on the role of C6 ceramide in the progression of canine mammary cancer and its potential mechanisms in the progression of this type of cancer in vitro. The article is original and relevant to the scientific community. It is well written and meets the proposed objectives. It opens up avenues for future clinical trials in bitches with breast tumors. It is also important for comparative concology.
The methodology is appropriate, the experimental design is well-designed and it is anchored in appropriate bibliographical references. I consider the figures to be adequate.
The study is well laid out, well written and very interesting with potential application in the future.
In the abstract they should put a short sentence about what C6 ceramide is.
In the introduction, the authors should clarify the research into EGR3 as a target gene in this context of treatment with C6 ceramide in vitro 88
I congratulate the authors on their work; I believe it should be accepted.
Reviewer 3 Report
Comments and Suggestions for Authors
The article entitled “C6 Ceramide Inhibits Canine Mammary Cancer Growth and 2 Metastasis by Targeting EGR3 Through JAK1/STAT3 Signaling” is a well-written and well-designed paper aiming to identify a new biomarker for canine mammary gland tumors. The authors have performed several experiments, including, cell growth, migration, invasion, and western blot. Please, see my specific comments below:
1. At the end of the introduction section, there is no clear manuscript goal. Please, include.
2. Methods section, cell, and culture -> Please provide a validation statement and where the cell lines were investigated against Mycoplasma infection.
3. When providing a reagents subheading, all regents should be mentioned.
4. 2.5 subheading is too short. This subheading could mix with the next (2.6).
5. The experimental groups are not clear. The authors could provide an experimental design at the beginning of methods to clear groups. Are only two groups assessed? Treated or not with C6 ceramide?
6. How about the negative flow cytometry controls?
7. Some parts of the text seem to have different font sizes. Please, check.
8. The first description of the result methods is not informative. Please, exclude (lines 210-212).
9. Figure 2, migration images are terrible. Is not possible to see anything in A and C.
10. Figure 5 I -> also in very bad resolution and difficult to see anything.
Reviewer 4 Report
Comments and Suggestions for Authors
General comment: The authors presented an original work addressing the effects of C6 ceramide in canine mammary tumors. For this, the authors performed bot in vitro and in vivo studies.
The writing style should be improved.
Title: The title is short, concise, and adequate.
Abstract and keywords: The abstract is complete. The keywords should be different from those used in the tile.
Introduction: The authors provide a complete overview of the thematic. The aim was clearly stated in this section.
Materials and methods: They are properly described.
Results: They are clearly presented and described. The Results are supported by Figures.
The quality of Figures 1C, 2A, 2C, 2E, 5H and 5I should be imperatively improved.
Discussion: The Results are clearly discussed in this section.
Conclusion: The conclusions are supported by the Results.
Round 2
Reviewer 1 Report
Comments and Suggestions for Authors
Thank you for addressing my points. C6 is an interesting approach, worth further investigation.
Reviewer 3 Report
Comments and Suggestions for Authors
I have no further comments.